# Comprehensive Genomic Characterization of *Staphylococcus aureus* Isolated from Atopic Dermatitis Patients in Japan: Correlations with Disease Severity, Eruption Type, and Anatomical Site

Shoko Obata,[a] Junzo Hisatsune,[b,c,d] Hiroshi Kawasaki,[a,e,f] Ayano Fukushima-Nomura,[a] Tamotsu Ebihara,[a] Chika Arai,[c,d] Kanako Masuda,[c,d] Shoko Kutsuno,[b,c] Yasuhisa Iwao,[b] Motoyuki Sugai,[b,c,d] Masayuki Amagai,[a,e] Keiji Tanese[a]

[a]Department of Dermatology, Keio University School of Medicine, Tokyo, Japan
[b]Antimicrobial Resistance Research Center, National Institute of Infectious Diseases, Tokyo, Japan
[c]Department of Antimicrobial Resistance, Graduate School of Biomedical and Health Sciences, Hiroshima University, Hiroshima, Japan
[d]Project Research Center for Nosocomial Infectious Diseases, Hiroshima University, Hiroshima, Japan
[e]Laboratory for Skin Homeostasis, RIKEN Center for Integrative Medical Sciences, Yokohama, Japan
[f]Laboratory for Developmental Genetics, RIKEN Center for Integrative Medical Sciences, Yokohama, Japan

Shoko Obata and Junzo Hisatsune contributed equally to this work. Author order was determined in Japanese alphabetical order.

**ABSTRACT** Atopic dermatitis (AD) shows frequent recurrence. *Staphylococcus aureus* is the primary microbial component in AD and is associated with disease activity. However, traditional typing methods have failed to characterize virulent AD isolates at the clone level. We conducted a comprehensive genomic characterization of *S. aureus* strains isolated from the skin of AD patients and healthy donors, comparing the whole-genome sequences of the 261 isolates with anatomical and lesional (AD-A)/nonlesional (AD-NL)/healthy sites, eruption types, clinical scores, virulence, and antimicrobial resistance gene repertoires in Japan. Sequence type (ST) diversity was lost with worsening disease activity; ST188 was the most frequently detected ST in AD-A and had the strongest correlation with AD according to the culture rate and proportion with worsening disease activity. ST188 and ST20 isolates inhabited all skin conditions, with significantly higher proportions in AD skin than in healthy skin. ST8, ST15, and ST5 proportions were equivalent for all skin conditions; ST30 was detected only in healthy skin; and ST12 was detected only in AD skin. ST97 detected in AD-A and healthy skin was clearly branched into two subclades, designated ST97$_A$ and ST97$_H$. A comparison of two genomes led to the discovery that only ST97$_A$ possessed the complete *trp* operon, enabling bacterial survival without exogenous tryptophan (Trp) on AD skin, where the Trp level was significantly reduced. Primary STs showing an AD skin inhabitation trend (ST188, ST97$_A$, ST20, and ST12) were all *trp* operon positive. The predominant clones (ST188 and ST97) possessed almost no enterotoxin genes, no *mecA* gene, and few other antimicrobial resistance genes, different from the trend observed in Europe/North America.

**IMPORTANCE** While *Staphylococcus aureus* is a member of the normal human skin flora, its strong association with the onset of atopic dermatitis (AD) has been suggested. However, previous studies failed to assign specific clones relevant to disease activities. Enterotoxins produced by *S. aureus* have been suggested to aggravate and exacerbate the inflammation of AD skin, but their role remains ambiguous. We conducted a nuanced comprehensive characterization of isolates from AD patients and healthy donors, comparing the whole-genome sequences of the isolates with anatomical and lesional/nonlesional/healthy sites, eruption types, clinical scores, virulence, and antimicrobial resistance gene repertoires in Japan. We demonstrate that specific clones are associated with disease severity and clinical manifestations, and the dominant clones are devoid of enterotoxin genes

Address correspondence to Keiji Tanese, tanese@a3.keio.jp, or Motoyuki Sugai, sugai@niid.go.jp.

The authors declare no conflict of interest.

and antimicrobial resistance genes. These findings undermine the established notion of the pathophysiological function of *S. aureus* associated with AD and introduce a new concept of *S. aureus* colonization in AD.

**KEYWORDS** atopic dermatitis, glabella, back, cubital fossa, volar forearm, SCORAD, *Staphylococcus aureus*, whole-genome sequence, phylogenetic analysis, multilocus sequence typing, MLST, ST188, ST97, ST8

Atopic dermatitis (AD) is a common chronic inflammatory skin disease that recurs frequently. The disease manifests with various symptoms such as an itchy rash, dryness, and eczema. Several factors have been suggested for the pathogenesis of AD, including genetic susceptibility, an activated type 2 immune system, an impaired skin barrier, and environmental factors (1). Additionally, increasing evidence has highlighted the role of skin microbiome dysbiosis in AD (2).

In particular, the proportion of *Staphylococcus aureus* bacteria in the skin flora is higher in patients with AD than in healthy subjects, suggesting an association between *S. aureus* colonization and worsening skin symptoms (3, 4). Whether *S. aureus* plays a significant role in the initiation of AD or acts secondary to the onset of AD remains unknown. A recent shotgun metagenomic sequence analysis of skin samples from a cohort of pediatric AD patients to study temporal dynamics throughout the disease course suggested that *S. aureus* is a primary driver of the disease at the species level (2). *S. aureus* is usually an asymptomatic inhabitant of the skin. However, when typical cutaneous structures or physiological conditions are disrupted, *S. aureus* can cause various infections by producing toxins, enzymes, and other virulence factors (5, 6). The virulence of *S. aureus* and its ability to produce pathogenic factors vary among its strains (7). Therefore, to evaluate the pathogenicity of *S. aureus* in specific cutaneous lesions, it is necessary to characterize strains cultured from related sites.

Multilocus sequence typing (MLST) has been used to identify pathogenic *S. aureus* strains in AD skin and to examine regional differences in the distributions of *S. aureus* strains (8–11), which has attracted interest in the role of clone-level differences in AD etiology and pathogenesis (12). However, most studies employed only MLST and PCR detection for virulence and antimicrobial resistance genes (ARGs). Nevertheless, they did not conduct a comprehensive characterization of *S. aureus* virulence at the clone level. Moreover, differences in genotypes according to anatomical region, disease severity, and eruption type have not been sufficiently investigated. In the present study, we performed swab sampling at four predetermined skin sampling sites of healthy donors and AD patients to culture *S. aureus*. In addition, we conducted a comprehensive genomic characterization of all of the isolates using whole-genome sequencing (WGS) analysis to determine clone-level correlations with various clinical/bacteriological data.

## RESULTS

**Patient characteristics.** A schematic flowchart of the swab sampling, culture, and genotyping procedures is summarized in Fig. S1 in the supplemental material. The characteristics of patients with AD and healthy skin are summarized in Table 1 and Table S1, respectively. The mean age of the 145 AD patients was 37.2 ± 11.6 years, and 53.1% (77/145) were male. The mean scoring AD (SCORAD) score was 27.1 ± 15.4. Of the 579 sites sampled, 50.9% had active lesions. In addition, active lesions were noted in 28.8% (85/295), 34.6% (102/295), 16.6% (49/295), and 20.0% (59/295) of the glabella, back, cubital fossa, and volar forearm samples, respectively (Table 1). Seventy-two healthy volunteers were enrolled in this study and were found to have no skin diseases at the time of enrollment, as assessed via examination by dermatologists. Samples from healthy skin were collected from either a 4- or a 25-cm² area (Table S1).

**S. aureus culture rate.** The culture rate of *S. aureus* collected from a 4-cm² skin area was 57.9% (84/145) in 145 AD patients, versus 5.6% (2/36) in 36 healthy individuals ($P < 0.001$). Since *S. aureus* was barely cultured in specimens obtained from the 4-cm² skin area of healthy individuals, we expanded the skin area to 25 cm² in healthy individuals to maintain a constant sample volume to determine the genotypes (Fig. S1). In total, 267

**TABLE 1** Demographics of the patients and lesions (579 lesions in 145 cases)

| Factor | Value for all patients (n = 145) |
|---|---|
| Mean age (yrs) ± SD (range) | 37.2 ± 11.6 (16–75) |
| No. (%) of patients of sex | |
| Male | 77 (53) |
| Female | 68 (47) |
| Mean SCORAD score ± SD (range)[a] | 27.1 ± 15.4 (1.4–70.5) |
| No. (%) of patients with SCORAD score of: | |
| <25 | 73 (53) |
| ≥25, <50 | 48 (35) |
| ≥50 | 16 (12) |
| No. (%) of samples from evaluation site | 579 |
| Nonlesional skin | 284 |
| Active lesion | 295 |
| Glabella[a] | 85 (29) |
| Erythema | 81 |
| Induration/papulation | 13 |
| Excoriation | 5 |
| Lichenification | 22 |
| Back[a] | 102 (35) |
| Erythema | 68 |
| Induration/papulation | 44 |
| Excoriation | 31 |
| Lichenification | 28 |
| Cubital fossa[a] | 49 (16) |
| Erythema | 39 |
| Induration/papulation | 13 |
| Excoriation | 5 |
| Lichenification | 22 |
| Volar forearm[b] | 59 (20) |
| Erythema | 31 |
| Induration/papulation | 37 |
| Excoriation | 14 |
| Lichenification | 13 |

[a]Evaluated in 137 patients.
[b]A single active lesion can contain multiple eruption types.

specimens were obtained from 25 cm$^2$ of skin from 67 healthy individuals. The characteristics of the study individuals are summarized in Table S1. *S. aureus* was cultured from 23.6% (63/267) of the sampling sites (Fig. S1). The culture rates in healthy skin, nonlesional atopic dermatitis skin (AD-NL), and atopic dermatitis skin with active lesions (AD-A) were 1.4% (2/144), 15.1% (43/284), and 52.5% (155/295), respectively (Fig. 1).

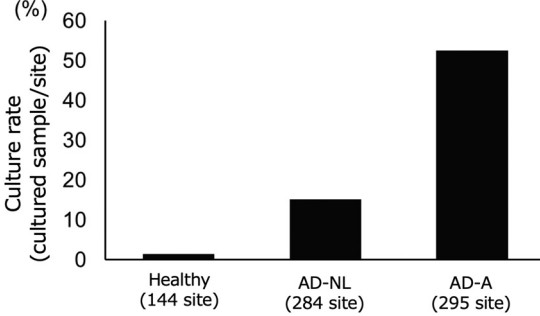

**FIG 1** Culture rates of *Staphylococcus aureus*. (A) Culture rates of *S. aureus* in atopic dermatitis (AD) and healthy skin. *S. aureus* was cultured from samples collected from a 4-cm$^2$ area at 579 sites in 145 AD patients and 144 sites in 36 healthy individuals. The culture rates in healthy skin, nonlesional atopic dermatitis skin (AD-NL), and atopic dermatitis skin with active lesions (AD-A) are demonstrated.

**TABLE 2** Multivariate logistic regression analysis of variables affecting *S. aureus* inhabitation of AD skin[a]

| Variable | Regression coefficient | SE | Odds ratio (95% CI) | P value |
|---|---|---|---|---|
| **Active AD lesions** | | | | |
| Eruption type | | | | |
| Erythema | 0.772 | 0.345 | 2.164 (1.102–4.252) | 0.025* |
| Induration/papulation | 0.926 | 0.324 | 2.525 (1.338–4.764) | 0.004* |
| Excoriation | 1.279 | 0.368 | 3.594 (1.745–7.400) | 0.001* |
| Lichenification | 0.468 | 0.308 | 1.597 (0.874–2.919) | 0.128 |
| Age | −0.010 | 0.012 | 0.99 (0.968–1.013) | 0.399 |
| Male sex | 0.686 | 0.274 | 1.986 (1.162–3.397) | 0.012* |
| Anatomical site | | | | |
| Cubital fossa | - | - | - | - |
| Volar forearm | 0.109 | 0.437 | 1.116 (0.474–2.628) | 0.802 |
| Glabella | 1.133 | 0.396 | 3.105 (1.429–6.749) | 0.004* |
| Back | −0.068 | 0.384 | 0.934 (0.44–1.982) | 0.860 |
| | | | | |
| **Nonlesional AD skin** | | | | |
| Age | 0.032 | 0.018 | 1.033 (0.997–1.070) | 0.070 |
| Male sex | 0.778 | 0.396 | 2.178 (1.002–4.773) | 0.049* |
| Anatomical site | | | | |
| Cubital fossa | - | - | - | - |
| Volar forearm | 0.404 | 0.489 | 1.498 (0.574–3.908) | 0.409 |
| Glabella | 1.549 | 0.498 | 4.708 (1.772–12.504) | 0.002* |
| Back | −0.109 | 0.673 | 0.897 (0.240–3.352) | 0.871 |
| SCORAD score | 0.048 | 0.014 | 1.050 (1.022–1.078) | 0.0004* |

[a]Anatomical sites, age, and sex were used as variables for both active AD lesion and AD nonlesional skin analyses. Eruption types were used as variables only for the analysis of AD active lesions, and the SCORAD score was used only for the analysis of AD nonlesional skin. The principal purpose of active AD lesion analysis was to identify trends of *S. aureus* inhabitation according to eruption types. The SCORAD scores were omitted because they were potentially not independent of the presence of eruptions. The purpose of nonlesional AD skin analysis, on the other hand, was to see whether *S. aureus* inhabitation is associated with systemic AD activity in locally nonlesional sites. CI, confidence interval. *, $P < 0.05$.

In the subgroup analysis according to the sampling site, the culture rates in samples from the glabella sites were 67.1% (57/85) in AD-A and 27.1% (16/59) in AD-NL, which were significantly higher than those at other sites ($P < 0.05$) (Fig. S2A). The culture rate of AD-A was higher in patients with SCORAD scores of ≥25 than in those with scores of <25 ($P < 0.001$). In addition, the culture rate in AD-NL increased with increasing SCORAD scores ($P < 0.001$) (Fig. S2B). Furthermore, the culture rate of lesions with excoriation was 67.3% (37/55), which was higher than that for other eruption types (Fig. S2C).

Multivariate logistic regression analysis of factors predisposing to *S. aureus* colonization in AD skin showed that the glabella site and lesions with erythema, excoriation, or papulation/induration were significant factors for AD-A (Table 2). In addition, the glabella site and AD disease severity were significant factors associated with AD-NL (Table 2). The culture rate of methicillin-resistant *S. aureus* (MRSA) in the total sample was 8% (21/261) (Table S4). The culture rates were 9% (14/155) for AD-A, 16.3% (7/43) for AD-NL, and 0% (0/63) for healthy skin.

**Phylogenetic overview of the 261 isolates.** A whole-genome-based phylogeny of 261 *S. aureus* draft genomes was reconstructed using kSNP3.0 to characterize the population structure of AD skin- or healthy skin-derived isolates in Japan (Fig. 2A). Using FastBAPS analysis, the maximum likelihood (ML) phylogenetic tree of 261 samples consisted of 13 sequence clusters (SCs). The Bayesian analysis of population structure (BAPS) and sequence type (ST) groups were well correlated with each other. The most dominant isolate SC was SC1, which comprised ST188 and ST1533 and the untyped STs belonging to clonal complex 188 (CC188), and SC7 comprised ST97 and SC6, which comprised ST8, ST999, ST2560, and ST380 (belonging to CC8). Contrary to reports in European countries (9–11), SC2, containing ST1 and ST2725 (belonging to CC1), was a minor cluster in Japan. The isolates from healthy donors formed clusters in the phylogenetic plot. Notably, all isolates of SC10, corresponding

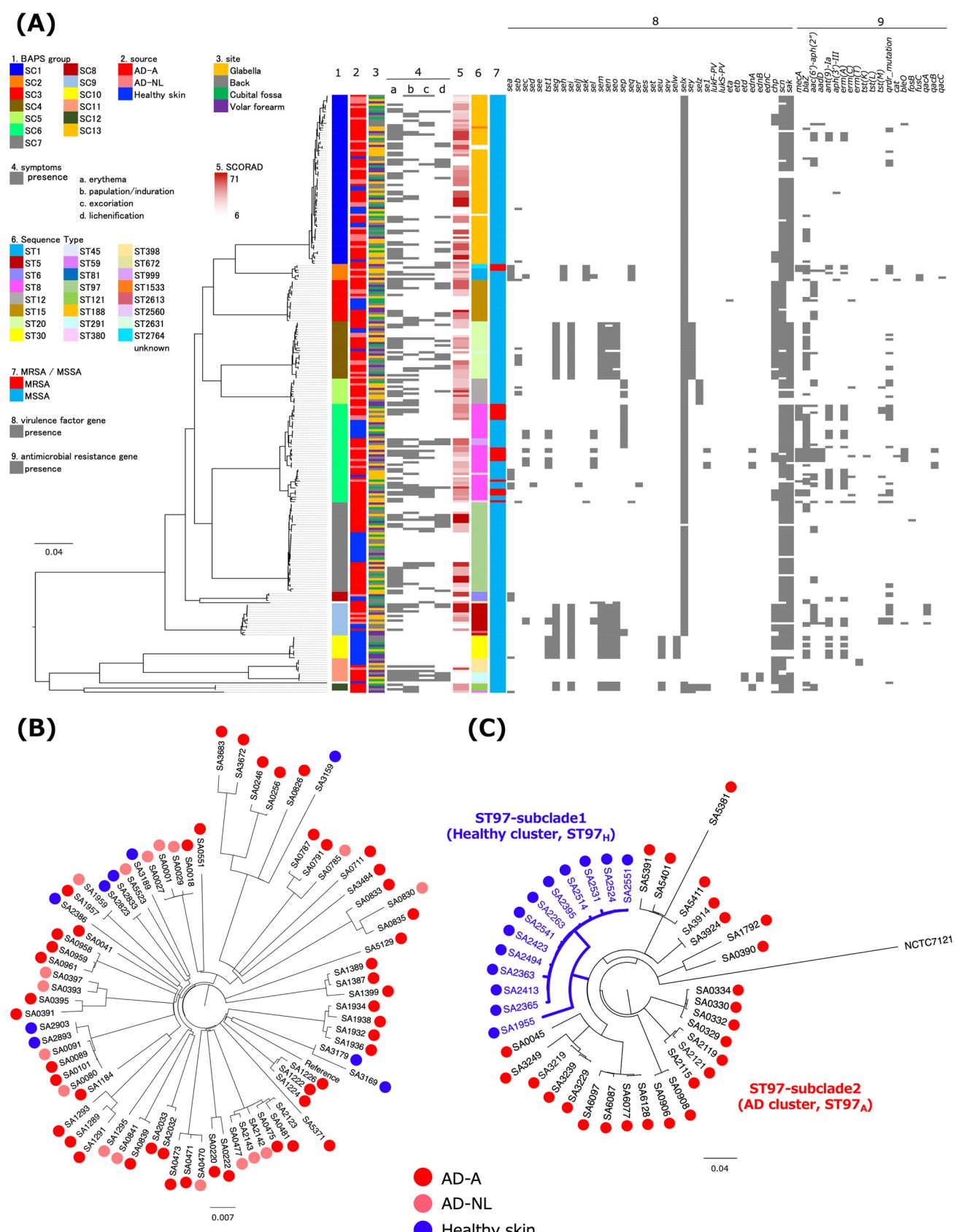

**FIG 2** Phylogenetic analysis and patterns of the presence of virulence factor genes (VFGs) or antimicrobial resistance genes (ARGs). (A, left) Maximum likelihood (ML) tree showing the phylogenetic structure of 261 *Staphylococcus aureus* isolates from atopic dermatitis (AD) or healthy skin. The ML tree

to ST30, were from healthy donors. In addition, some clusters of isolates from healthy donors were found in SC6, SC7, SC8, and SC9.

Among the 261 isolates, there were only 21 *mecA* gene-positive isolates; they were found in two clusters, SC2, corresponding to CC1 (ST1, $n = 1$; ST2764, $n = 2$), and SC6, corresponding to CC8 (ST8, $n = 17$; ST380, $n = 1$). All 21 MRSA isolates possessed the staphylococcal cassette chromosome *mec* type IV (SCC*mec* IV) gene (Table S4). Searching for the virulence genes of the isolates clearly showed that the most dominant clusters (SC1, SC3, and SC7) were devoid of superantigen genes, except for *selx*, which is almost ubiquitously present in *S. aureus* isolates from AD patients (Fig. 2).

**Characteristics of cultured *S. aureus* STs and their distribution in AD skin with active lesions, nonlesional AD skin, and healthy skin.** A single ST was identified at 132 sites among 295 sites in AD-A, 34 sites among 284 sites in AD-NL, and 61 sites among 267 sites in healthy skin (25 cm$^2$) (Fig. S3). Two STs were identified at 11 sites in AD-A, 4 sites in AD-NL, and 2 sites in healthy skin (25 cm$^2$). None of the sites contained more than three STs.

In both AD-A and AD-NL, the most frequently detected ST was ST188, followed by ST8 (Fig. 3A and Fig. S3B and C). In comparison, among healthy skin samples, the most frequently detected ST was ST97, followed by ST30.

The proportions of individual STs identified in AD-A, AD-NL, and 25-cm$^2$ samples of healthy skin were analyzed. ST188 and ST20 inhabited all skin conditions, with a significantly higher proportion in AD skin than in healthy skin ($P < 0.05$) (Fig. 3B). In contrast, the proportions of ST8, ST15, and ST5 were equivalent in healthy skin, AD-NL, and AD-A. In addition, ST30 was detected only in healthy skin, whereas ST12 was detected only in AD skin.

ST97 was detected in healthy skin and AD-A but not in AD-NL (Fig. 3). Phylogenetic analysis of the 261 isolates indicated that SC7 (ST97) was divided into clusters of healthy skin- and AD skin-derived isolates (Fig. 2A). Therefore, core-genome single nucleotide polymorphism (SNP) phylogenetic analysis was performed for 39 isolates in SC7 (ST97), and the results were compared with the data from the same analysis for SC1 (CC188), in which several healthy skin-derived isolates were scattered in the cluster. The SC7 (ST97) clade had a forked tree branch of ST97 subclade 1 (healthy cluster, ST97$_H$) and ST97 subclade 2 (AD cluster, ST97$_A$). However, such branching was not observed in the phylogeny plot of SC1 (CC188) (Fig. 2B and C). By comparative pangenome analysis, we identified the presence or absence of two operons (*trp* operon and *cst* operon) discriminating between the two clades of SC7 (ST97) (Fig. S4 and Table S5). (i) The tryptophan (Trp) biosynthetic pathway in *S. aureus* involves the *trp* operon that contains seven genes (*trpA*, *trpB*, *trpC*, *trpD*, *trpE*, *trpF*, and *trpG*) and is essential for growth (13). ST97$_A$ possesses all seven genes of the *trp* operon, whereas in ST97$_H$, four genes (*trpC*, *trpD*, *trpF*, and *trpG*) are lacking, and two genes (*trpB* and *trpE*) are truncated due to a recombination event between the direct repeats within these genes (Fig. S4B, top). When examining the boundary nucleotide sequences of the lacking DNA region in the *trp* operon of ST97$_A$, direct repeats of 10-bp sequences were found. (ii) ST97$_A$ possesses the sulfide stress resistance gene cluster consisting of the *cst* operon (*cstA*, *cstB*, and *sqr*), the persulfide-sensing transcriptional repressor gene *cstR*, and the sulfide exporter gene *tauE*, whereas ST97$_H$ does not possess this cluster (Fig. S4B, bottom). By examining the boundary nucleotide sequences of this gene cluster in ST97$_A$, a 468-bp duplicating sequence in the 3′ side of the tRNA-dihydrouridine synthase gene *dus* was found.

**Distribution of STs according to anatomical region, AD severity, and eruption type.** In the subgroup analyses according to the skin region sampled, the culture numbers of STs in the glabella, back, and cubital fossa sites were similar to those in the overall data. The

**FIG 2** Legend (Continued)

was constructed based on the whole-genome alignment of 139,973 SNP sites using kSNP3.0. The scale bar represents the number of nucleotide substitutions per site. In the heatmap, in column 1, the Bayesian analysis of population structure (BAPS) group refers to sequence clusters (SCs) identified by FastBAPS; in column 2, the sources are AD skin with active lesions (AD-A), nonlesional AD skin (AD-NL), and healthy skin; column 3 indicates the glabella, back, cubital fossa, and volar forearm sites; column 4 shows the presence/absence of erythema, papulation/induration, excoriation, and lichenification; column 5 shows the scoring atopic dermatitis (SCORAD) score; in column 6, each color represents a sequence type (ST); column 7 shows methicillin-resistant *S. aureus* (MRSA) or methicillin-susceptible *S. aureus* (MSSA); and in columns 8 and 9, the right matrix panel shows the presence/absence of VFGs and ARGs, with gray representing the genes' presence. (B) ML tree of 74 ST188 (belonging to SC1) isolates constructed based on the 5,595 SNP sites using the TVM or model and 100 bootstrap replicates with the JMUB1273 (ST188) reference genome. (C) ML tree of 39 ST97 (belonging to SC7) isolates constructed based on the 2,166 SNP sites using the TVM or model and 100 bootstrap replicates with the NCTC7121 (ST97) reference genome. The scale bar represents the number of nucleotide substitutions per site.

**(A)**

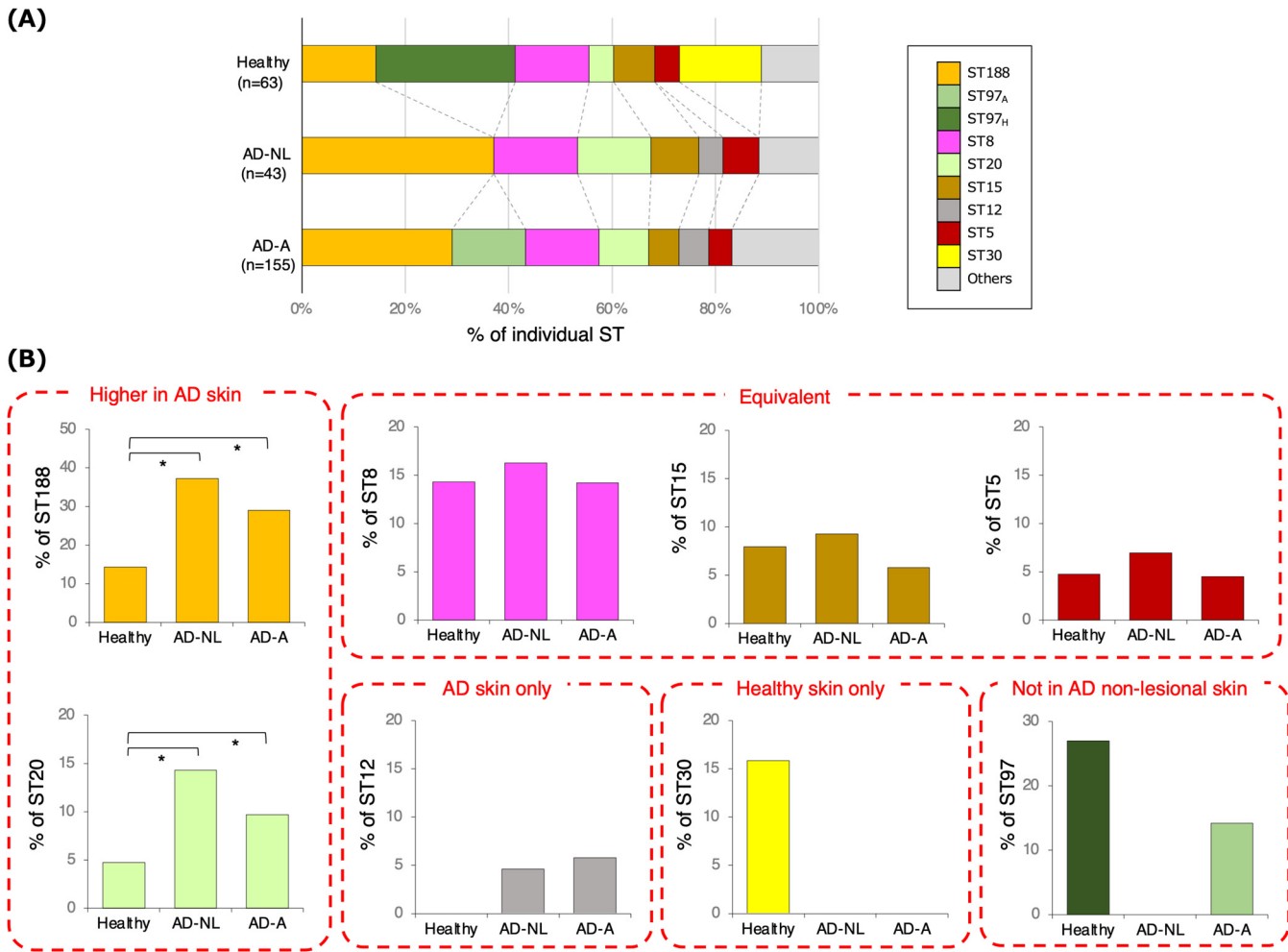

**FIG 3** Sequence types (STs) of cultured *Staphylococcus aureus* isolates. (A) The proportions of each ST among the total *S. aureus* genotypes, identified by multilocus sequence typing (MLST), in healthy skin, nonlesional atopic dermatitis skin, and atopic dermatitis skin with active lesions (AD-A) were calculated (63 for healthy skin, 42 for AD-NL, and 154 for AD-A). (B) MLST genotypes according to each skin condition. *$P < 0.05$.

proportions of ST188 in AD-A and ST97$_H$ in healthy skin were the highest among the glabella, back, and cubital fossa sites (Fig. 4). However, in the volar forearm site, the various STs showed equivalent proportions. The culture rate and proportion of ST188 in AD-A of the volar forearm site were approximately half of those of the other regions (Fig. S5B and C). ST12, which was not detected in healthy skin, was detected only in AD-NL at the glabella site (Fig. 4).

The SCORAD scores showed that the genotypic diversity of *S. aureus* decreased in both AD-A and AD-NL with worsening AD severity (Fig. 5A and Fig. S6A and B). In contrast, the culture rate and proportion of ST188 increased with worsening disease activity in both the AD-A and AD-NL groups. ST97$_A$ was observed only in AD-A, but it was the second most prevalent ST in patients with a SCORAD score of ≥50 (Fig. 5A). In contrast, ST8 was not detected in patients with a SCORAD score of ≥50 (Fig. 5B and Fig. S6C and D). In healthy individuals, the proportion of ST188 was 12.8% (Fig. 3B), lower than that in individuals with a SCORAD score of <25 (Fig. 5), suggesting that ST188 had the strongest correlation with AD disease activity.

In lesions with erythema or papulation/induration, the most frequently detected ST was ST188, followed by ST8, ST97$_A$, and ST20, similar to the trend observed in the overall sample set (Fig. 3A and Fig. 6). In contrast, ST97$_A$ and ST8 accounted for a higher proportion than ST188, whereas ST12 and ST5 were not detected in lesions with excoriation (Fig. 6). In lesions with lichenification, ST8 and ST12 accounted for lower proportions than those at the other sites.

**Characteristics of virulence factor gene or ARG patterns on AD skin with active lesions, nonlesional AD skin, and healthy skin.** Many virulence factor genes (VFGs) and ARGs are located on genomic islands and plasmids. These genes are acquired by

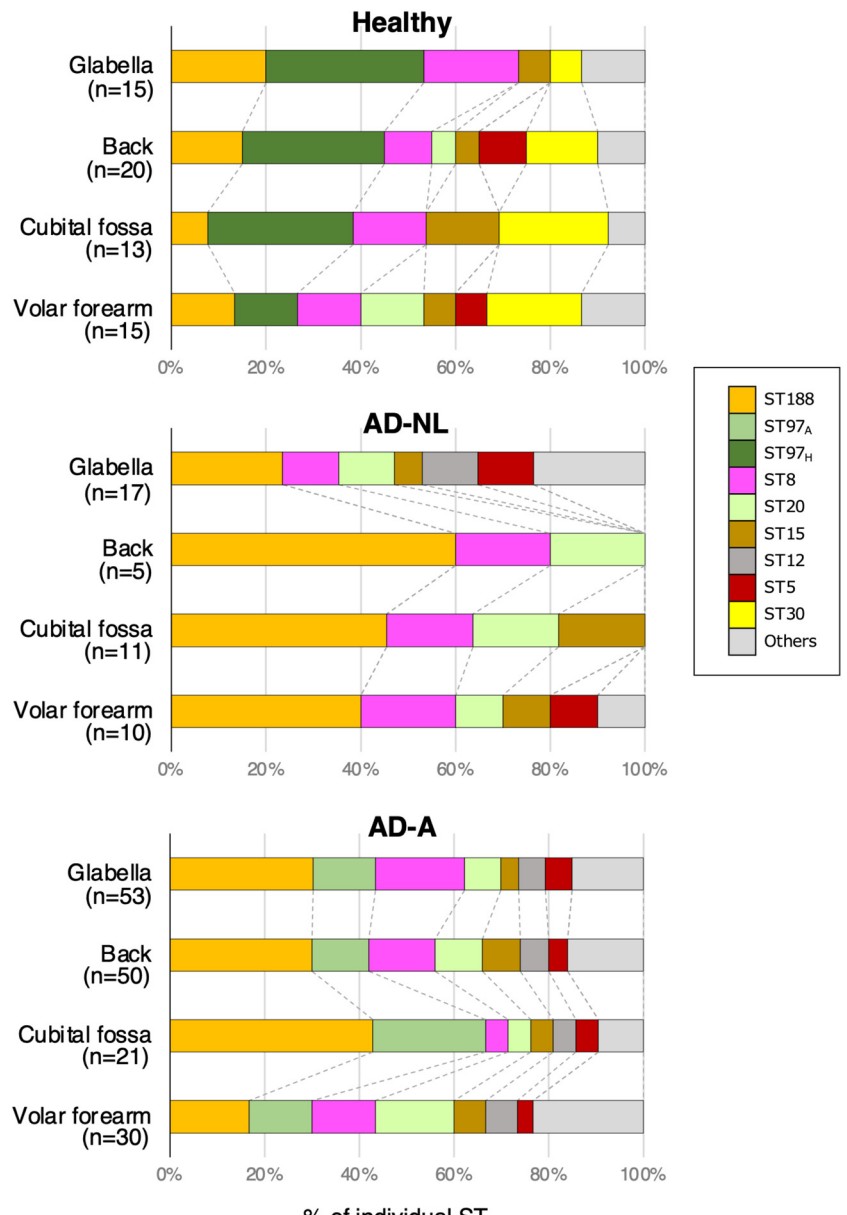

**FIG 4** Sequence types (STs) of cultured *Staphylococcus aureus* isolates according to sampling site. The proportions of STs among the total number of STs identified at each sampling site in healthy skin, nonlesional atopic dermatitis skin (AD-NL), and atopic dermatitis skin with active lesions (AD-A) are shown.

mobile genetic elements (MGEs) such as bacteriophages, transposons, or plasmids (14). First, the presence of VFGs in *S. aureus* isolates from AD or healthy skin was investigated. The proportions of most staphylococcal enterotoxin (SE), exfoliative toxin (ET), and epidermal cell differentiation inhibitor (EDIN) genes of AD skin- and healthy skin-derived isolates were 0 to 30%, except for the *selx* gene, and the prevalence patterns were similar (Fig. 7). *selx* is located in the core genome of *S. aureus* and is present in most strains but not in ST30, ST398, or ST291 AD skin-derived isolates (Table S2). The proportion of immune evasion clusters (IECs) (*chp*, *scn*, and *sak*) in AD skin-derived isolates was the same as that in the healthy skin-derived isolates (Fig. 7). No Panton-Valentine leukocidin (PVL) genes (*lukF-PV* and *lukS-PV*) were detected in AD skin- or healthy skin-derived isolates, and the prevalence of the ET and EDIN genes was extremely low. Among the dominant ST types (ST188, ST8, ST97, and ST30) of AD skin- or healthy skin-derived isolates, ST188 and ST97 possessed almost no

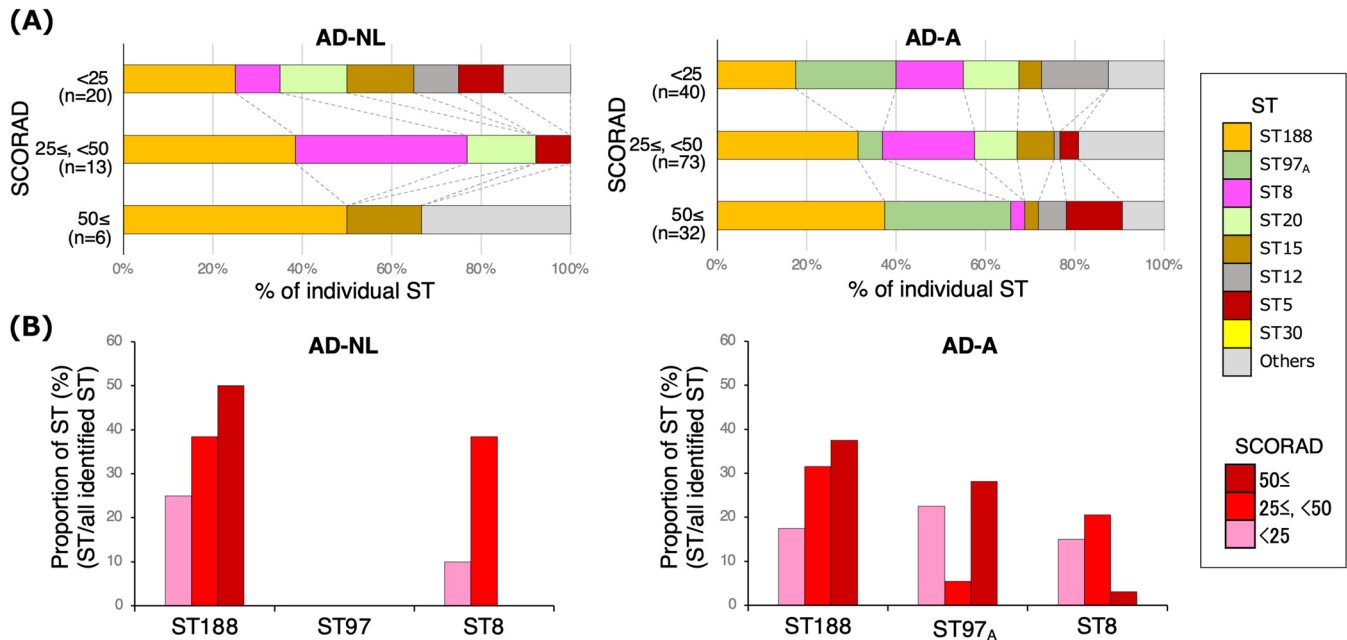

FIG 5 Sequence types (STs) of cultured *Staphylococcus aureus* isolates according to disease severity. (A) Proportion of each multilocus sequence typing (MLST) genotype among all identified MLST genotypes, according to disease severity, stratified according to the scoring atopic dermatitis (SCORAD) score in nonlesional atopic dermatitis skin (AD-NL) and atopic dermatitis skin with active lesions (AD-A). (B) Proportions of ST188, ST97$_A$, and ST8 in AD-NL and AD-A, according to disease severity.

VFGs except for *selx*, *scn*, and *sak* (Fig. S7). Only 22.2% of ST188 healthy skin-derived isolates carried *seb*, but ST188 AD skin-derived isolates did not (Fig. S7).

The rates of the existence of ARGs in AD skin- and healthy skin-derived isolates were compared (Fig. 7). The proportions of some ARGs in AD skin-derived isolates were higher than those in healthy skin-derived isolates: those of the β-lactam resistance gene *mecA* were 9% (14/155) in AD-A and 16.3% (7/43) in AD-NL, those of the aminoglycoside resistance gene *aac(6′)-aph(2″)* were 30.3% (47/155) in AD-A and 20.9% (9/43) in AD-NL, those of *aadD* were 3.2% (5/155) in AD-A and 9.3% (4/43) in AD-NL, those of the tetracycline resistance gene *tet*(M) were 3.9% (6/155) in AD-A and 7% (3/43) in AD-NL, and those of the quinolone resistance mutations were 21.3% (33/155) in AD-A and 25.6% (11/43) in AD-NL (Fig. 7). However, the proportions of the erythromycin resistance gene *erm*(A) and the aminoglycoside resistance gene *ant(9)-la* in healthy skin-derived isolates were both 25.4% (16/63), significantly higher than those in the AD skin-derived isolates (Fig. 7). Among the dominant STs of AD skin- or healthy skin-derived isolates, ST188, ST97$_H$, and ST30 had few ARGs (Fig. S7). AD skin-derived ST8 isolates had *mecA* (58.6%; 17/29), *aac*

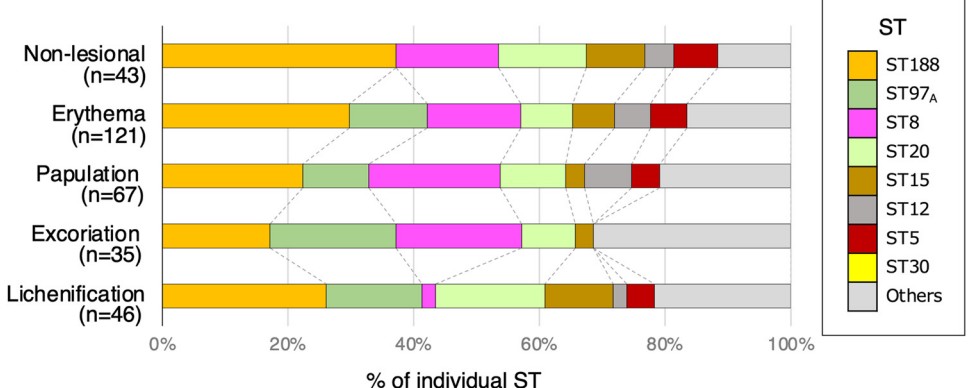

FIG 6 Sequence types (STs) of cultured *Staphylococcus aureus* isolates according to eruption type. The proportions of STs among the total identified STs according to eruption type are shown.

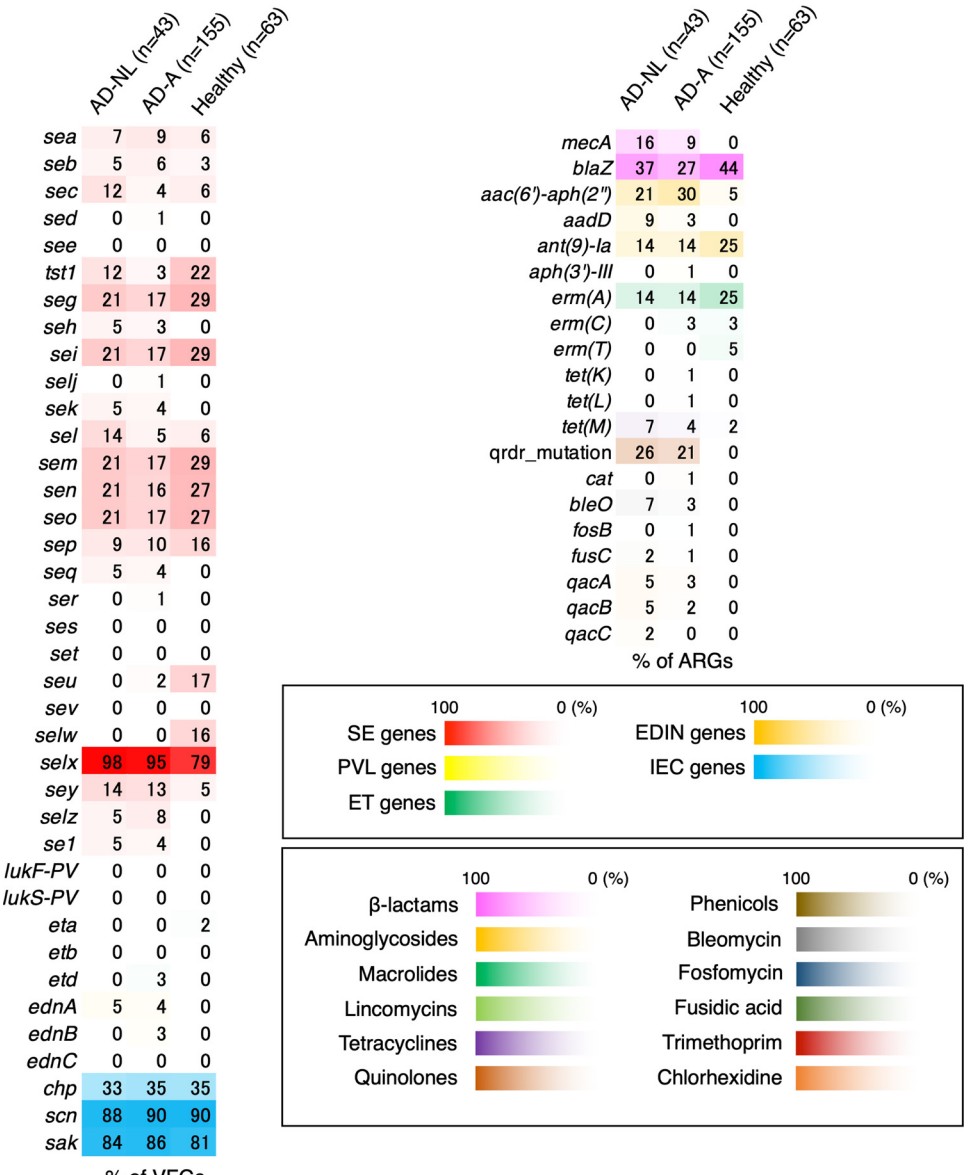

**FIG 7** Characteristics of virulence factor gene (VFG) or antimicrobial resistance gene (ARG) patterns of atopic dermatitis (AD) skin- or healthy skin-derived isolates. The proportions of several VFGs (left) and ARGs (right) in AD-NL, AD-A, and healthy skin are presented as heatmaps. The numbers in the heatmap columns are percentages.

(6')-aph(2″) (37.9%; 11/29), *aadD* (20.7%; 6/29), quinolone resistance mutations (41.4%; 12/29), the bleomycin resistance gene *bleO* (20.7%; 6/29), and the disinfectant resistance gene *qacB* (17.2%; 5/29); these proportions were significantly higher than those in healthy skin-derived ST8 isolates (Fig. S7).

## DISCUSSION

Bacteria and the host skin environment are involved in a variety of interactions, and specific populations of *S. aureus* may have a higher affinity for the skin environment owing to skin disorders. AD is classically linked to a stronger Th2 response and more *S. aureus* colonization, whereas psoriasis is linked to a stronger Th1 response and less *S. aureus* colonization (15). This may be due to immunological/priming profiles and not strain variation/virulence factor profiles. Additionally, a Th1/Th17 response is important for the elimination of *S. aureus* from the skin (16).

In the present study, the rate of colonization by *S. aureus* in AD patients (AD-NL) was 21.4% (31/145), and the infection rate (AD-A) was 51.7% (75/145), both high, consistent with previous reports (3, 4). In addition, we investigated the rate of detection of a single clone or multiple clones (including two or more clones) in each AD patient and found that for a single clone, the rate was 48.3% (70/145), and for multiple clones, the rate was 10.3% (15/145). Moreover, the culture rate was significantly higher in the AD-A group than in the AD-NL group and in patients with higher SCORAD scores. The glabella site had a significantly higher culture rate among the four sampling sites in both AD and healthy skin. Logistic regression analysis of AD-NL revealed that the glabella site and the SCORAD score significantly predicted *S. aureus* inhabitation. These results suggest that seborrheic sites and underlying AD disease activity are host-related factors that predispose individuals to *S. aureus* inhabitation, even without clinically evident dermatitis.

Phylogenetic analysis and the identification of specific ST lineages of *S. aureus* using WGS revealed different skin inhabitation trends among the primary STs. ST188 and ST20 were found in both skin types but were more common in AD skin. ST8, ST15, and ST5 were found in equal proportions in both skin types. ST30 and ST12 were found in healthy and AD skin, respectively. $ST97_H$ was found in healthy skin, and $ST97_A$ was found in AD-A.

ST188 was the most frequently detected ST in AD skin, and its culture rate and proportion increased significantly with worsening disease activity. ST188 is also the most frequently detected ST in Asian patients with AD and infants before AD onset (17, 18). ST188, a double-locus variant of ST1 (19), may play a significant role in the pathogenesis of the Japanese AD skin microbiome instead of ST1. ST20 was one of the top five STs detected at all of the sampling sites and showed a proportion similar to that of ST188, which was higher in AD skin. ST20 is rich in VFGs, including several SE genes, whereas ST188 possesses few VFGs. ST20 was preferentially present in the volar forearm site in AD-A among the four sampling sites, whereas ST188 was predominantly prevalent in the glabella, back, and cubital fossa sites but not in the volar forearm site in AD-A. ST20 is more likely associated with disease activity in Japanese patients with AD. ST12 was not detected in healthy skin but was detected only in AD-NL at the glabella site. However, it was also detected in all four sampling sites in AD-A, suggesting that its colonization is associated with disease activity. These results suggest that *S. aureus* strains comprising the skin flora of the extremities differ from those of the back, cubital fossa, and glabella sites in healthy and AD skin samples.

ST8, including its hypervirulent clone ST8-MRSA USA300, is found worldwide (20). However, USA300 was not identified among the 17 ST8-MRSA clones in the present study. ST15 is one of the most common clones in Asian and African countries and is detected in diseased individuals and asymptomatic carriers (21, 22). ST5 is also found worldwide and has many variants, including epidemic-causing MRSA clones such as New York/Japan and USA100 (23).

This study also identified STs found exclusively in normal and AD skin: ST30 and ST12, respectively. As the most frequently detected genotype in healthy skin of European and North American populations is CC30 (CC containing ST30) (11, 24, 25), ST30 may inhabit healthy skin regardless of region or ethnicity. Although the number of ST12 clones identified was insignificant, they were detected more frequently in AD-A than in AD-NL, which may be related to AD pathogenesis. ST12 is an important methicillin-susceptible *S. aureus* lineage and is significantly associated with infectious endocarditis (26, 27). However, there is no evidence of an association between ST12 and AD activity. Characterization of ST12 may clarify the properties of *S. aureus* that exacerbate AD symptoms.

$ST97_A$ possesses a complete *trp* operon, which plays a vital role in Trp biosynthesis via the anthranilate pathway (13), while $ST97_H$ has a truncating mutation in the operon, suggesting that $ST97_A$ is able to produce Trp independently and that $ST97_H$ is not.

It has been demonstrated that the levels of Trp and its metabolites are significantly reduced on AD skin; therefore, *S. aureus* does not require exogenous Trp to selectively grow on AD skin (28). Indeed, Fyhrquist et al. reported that the skin of patients with AD is preferentially colonized by those strains that are able to synthesize Trp, according to microbiome analysis (29). The microbes are supposed to degrade intrinsically produced Trp, and the

generated metabolite 3-hydroxyanthranilic acid (3-HAA) presumably acts as an inflammatory mediator. We speculate that *trp* operon-positive ST97$_A$ is capable of persistent colonization in active AD lesions but that *trp* operon-negative ST97$_H$ is not. By analogy, the isolates belonging to ST188, ST20, and ST12, which were found to be relevant to AD skin, were all positive for the *trp* operon. However, all of the isolates of ST30, which is present only on the skin of healthy donors but not on AD skin, were also positive for the *trp* operon. Nevertheless, the detailed mechanism delineating their location in healthy skin and AD skin remains to be further clarified.

Additionally, ST97$_A$ carries the sulfide stress resistance gene cluster, which may have been recombined at a 468-bp duplicated sequence within the *dus* gene. The *cst* operon (*cstA*, *cstB*, and *sqr*) is vital for the survival of *S. aureus* under hydrogen sulfide (H$_2$S) stress (30). The relationship between H$_2$S and human skin diseases was recently reported by Xiao et al. (31). The aberrant metabolism of H$_2$S is involved in the pathogenesis of several skin diseases such as angiopathy, psoriasis, ulcers, pigmentary disorders, and melanoma (31). However, the effects of *S. aureus* on AD and H$_2$S are unknown.

The most dominant clones in AD, ST188 and ST97$_A$, were devoid of enterotoxin genes except for *selx*. They were all devoid of *mecA*, and there was no MRSA among the isolates (Fig. 2A; see also Fig. S7 in the supplemental material). However, these isolates did not possess any CRISPR-Cas system or peculiar restriction-modification systems of the characteristic DNA elements, as determined by pangenome analysis (Fig. S4A). The reason why dominant isolates from AD skin lack most of the enterotoxin genes and the *mecA* gene remains unknown.

Based on the discussion above, strains inhabiting AD skin at higher proportions, such as ST188 and ST97$_A$ strains, may be closely associated with AD disease severity. However, the affinity of each strain may vary depending on the host skin environment, and each strain may have a different pathogenic role in individual AD lesions. The STs identified from the volar forearm site showed trends different from those of the STs identified from the other sampling sites, and ST diversity was lost with worsening disease activity. In addition, the STs identified from lesions with excoriation showed trends different from those of the STs identified from the other eruption types. Further investigation is necessary to clarify the pathogenic roles of the various STs.

Several studies from Europe and North America have reported that CC1 (CC including ST1) is the most enriched clone in AD skin (9, 11, 32). We tried to compare the genomic characteristics of AD skin-derived isolates from Japan and Europe/North America to determine the regional differences in AD skin-derived isolates. A data set of AD skin-derived isolates deduced using WGS was searched for in a public database. One data set from Denmark was available for comparison with the data in this study. Therefore, the data set containing the data for AD-A-, AD-NL-, or healthy skin-derived isolates obtained using WGS was used (33). In the data set from the Denmark study, the predominant clone in AD-A-derived isolates was ST1, which was distinct from that in Japan (Table S2). CC188 was different from CC1 in two of the seven alleles from MLST. In this study, the proportions of VFGs and ARGs in the predominant STs of AD skin-derived isolates were examined and compared with those in the Danish data set (Fig. 8). Japanese AD-dominant STs (ST188 and ST97) had significantly lower proportions of VFGs and ARGs, as described above. In contrast, the Danish AD-dominant ST types, ST1 and ST45, carried several SE genes: ST1 carried *sea*, *sek*, *sep*, and *selx*, and ST45 carried *sec*, *seg*, *sei*, *sel*, *sem*, *sen*, *seo*, *selw*, and *selx* (Fig. 8). Most of the SE genes were shown to encode superantigens with proinflammatory properties (34). These properties may cause exacerbations of AD and affect the clinical manifestations of AD. Furthermore, ST1 possessed *aac(6′)-aph(2″)* (58.8%) and the fusidic acid resistance gene *fusC* (94.1%). ST45 possessed quinolone resistance mutations (100%). A higher proportion of resistance gene-positive isolates may decrease antimicrobial treatment options.

This study has several limitations. First, this was a retrospective study; therefore, the assessments of the evaluators were not standardized. Second, although this study involved

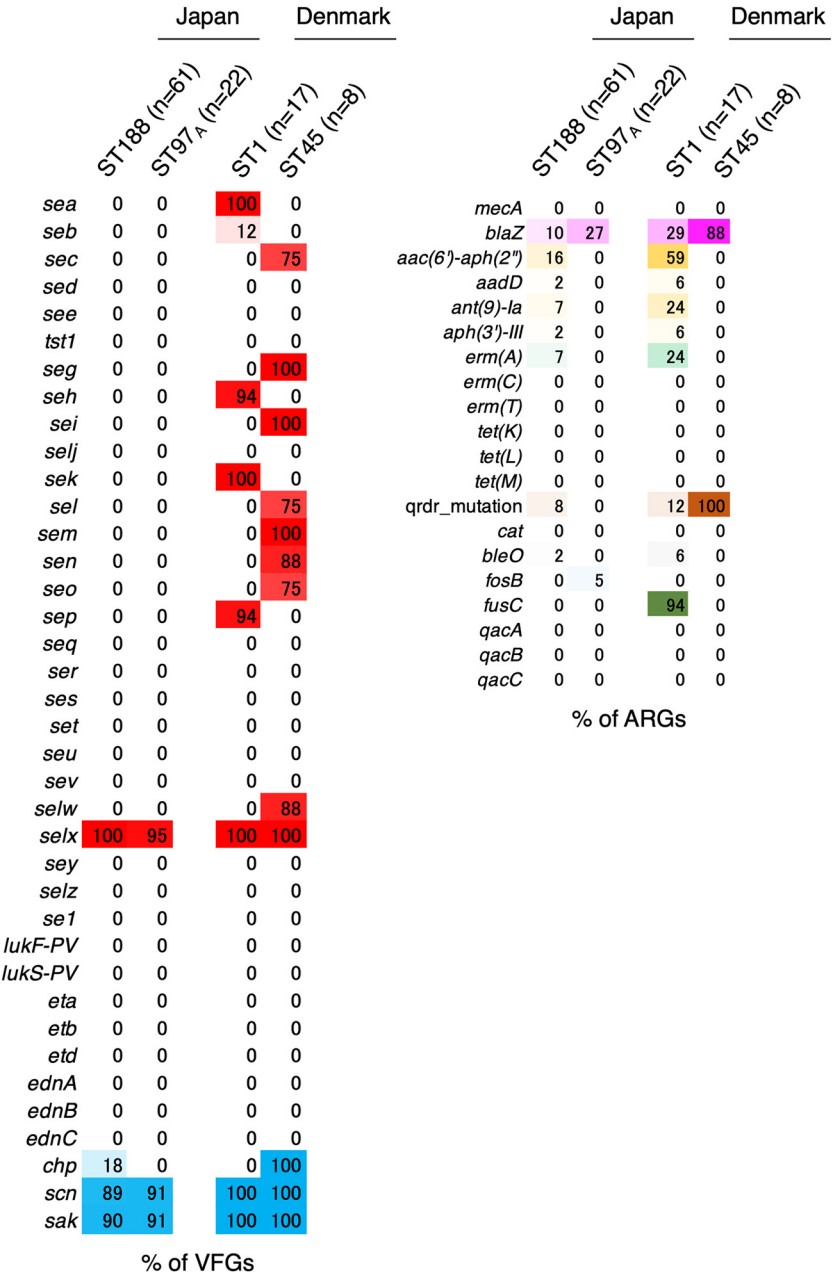

**FIG 8** Comparative analysis of VFG or ARG patterns of atopic dermatitis skin with active lesions (AD-A)- and nonlesional atopic dermatitis skin (AD-NL)-derived isolates from Japan or Denmark (BioProject accession no. PRJEB18560). Heatmaps show VFGs or ARGs of predominant STs in Japan or Denmark. AD_ls, active AD lesion; AD_nls, nonactive AD lesion. The proportions of several VFGs (left) and ARGs (right) are shown, and heatmap colors represent each category.

sampling from many assessment sites, several subpopulations did not have sufficient samples for analysis.

In conclusion, *S. aureus* is more likely to inhabit AD skin than healthy skin and AD-A than AD-NL. ST188 accounted for a high proportion of all *S. aureus* STs identified in AD skin, and the ST188 proportion is correlated with AD disease activity. However, the distribution of each ST varies depending on the anatomical site, disease severity, and eruption type, which may affect individual AD lesions differently. Finally, there was a clonal difference in the significant strains isolated from patients with AD. The critical difference in the numbers of VFGs and ARGs in the major clones may reflect the clinical manifestations of the *S. aureus*-involved pathogenesis of AD in Asia and Europe.

## MATERIALS AND METHODS

**Study participants.** This study evaluated patients with AD, aged 16 years and older, who visited Keio University Hospital between July 2016 and June 2020. The protocol was approved by the Keio University School of Medicine Ethics Committee (approval no. 20130384) and was performed in accordance with the Declaration of Helsinki. Written informed consent was obtained from all participants. AD was diagnosed according to the diagnostic criteria described previously by Hanifin and Rajka (35). The patients received standard treatments for AD, including emollients, topical corticosteroids, tacrolimus, oral antihistamines, and cyclosporine. Patients who had used antibiotics within the preceding week were excluded from the study.

**Sampling sites and methods for assessing cutaneous findings.** Disease activity was assessed using the SCORAD index (score range from 0 to 103) (36). *S. aureus* culture samples were collected from four predetermined sampling sites, which differed in sebum secretion and susceptibility to dermatitis, glabella, back, cubital fossa, and volar forearm, in patients with AD. Samples were evaluated for erythema, papulation/induration, excoriation, and lichenification using the eczema area and the severity index (37). Sampling sites with eruptions were defined as AD-A, whereas those without eruptions were defined as AD-NL. Disease activity was assessed by two dermatologists (H. Kawasaki and T.E.).

**Isolation of *S. aureus* from skin swab samples.** Samples for *S. aureus* culture were collected by rubbing the sampling sites with a sterile swab (catalog no. S0475-1; Osaki Medical Corporation, Nagoya, Japan) premoistened with sterile phosphate-buffered saline (PBS) (catalog no. D9PBS WX-3009-1980-9; As One, Osaka, Japan).

First, samples from AD and healthy skin within a 4-cm$^2$ frame were collected using a swab, and the culture rates of *S. aureus* were compared. Since specimens obtained from healthy skin (glabella, back, cubital fossa, and volar forearm) did not show significant growth of *S. aureus* in culture, the sampling area was expanded to 25 cm$^2$ in healthy controls to maintain a constant sample volume for genotype determination (see Fig. S1 in the supplemental material). Next, the bacterial isolates were cultured on mannitol salt agar (Nissui Pharmaceutical Co., Ltd., Tokyo, Japan). Individual colonies were selected, and *S. aureus* was identified by PCR of the *S. aureus* *femA* and *femB* genes (38).

**Genome sequencing, assembly, annotation, and pangenome analyses.** Genomic DNA was extracted using lysostaphin and the Agencourt AMPure XP system (Beckman Coulter, Inc., Brea, CA, USA), according to the manufacturer's instructions. DNA libraries were prepared for sequencing using Enzymatic 5X WGS reagents (BioStream Co., Ltd., Tokyo, Japan) (39) and the Biomek i7 Workstation automated sample preparation system (Beckman Coulter, Inc.). DNA sequencing was performed on the Illumina HiSeq X Five platform to generate 150-bp paired-end reads (Macrogen Japan Corporation, Tokyo, Japan). Raw reads were assembled using Shovill v1.0.9 (available at https://github.com/tseemann/shovill) with default settings. In addition, MLST was performed to define CCs and STs using mlst v2.19.0 (available at https://github.com/tseemann/mlst). The presence of VFGs (Table S3) of *S. aureus* was analyzed using a BLASTN search (40) against the assembled genome sequences, and the presence/absence decision was made based on thresholds of a sequence identity of ≥95% and a coverage of ≥80%. The presence and mutation of ARGs were detected using ResFinder 4.1 (41) with the ResFinder 2022-05-24 database and the PointFinder 2021-02-01 database with default parameters. Genomes were annotated using Prokka v1.14.6 (42). Pangenome analysis was performed using Roary v3.13.0 (43) with the parameter "-i 90 cd 99s." Nucleotide sequences were aligned using MAFFT (44).

**Phylogenetic and clustering analyses.** In the first step, the phylogenetic tree of the core-genome alignment was constructed using the kSNP3.0 program (45) without reference genomes. Next, genetic population structure analysis was performed by partitioning the strains into SCs of genetically similar individuals using the FastBAPS package v1.0.8 Bayesian hierarchical clustering program (46) in R v4.2.2 (47). Next, SNPs were detected in the core-genome-length alignment for each sequence cluster using Snippy v4.6.0 (available at https://github.com/tseemann/snippy) and mapped to a reference genome. Next, recombination-free aligned sequences were generated using Gubbins v3.2.0 (48) for each sequence cluster. The phylogenetic tree of recombination-free sequences was constructed using RAxML-NG v1.0.1 (49) with the best model inferred by ModelTest-NG v0.1.7 (50) and 100 bootstrap replicates. Finally, sequence comparisons and visualization of the results were performed using Phandango (51), FigTree v1.4.4 (available at http://tree.bio.ed.ac.uk/software/figtree/), and Easyfig v2.2.2 (52).

**Indicators used to assess *S. aureus* inhabitation.** First, the culture number was defined as the number of samples with *S. aureus* or a specific ST detected at the sample collection sites. The culture rate was defined as the percentage of *S. aureus* or a specific ST detected at the sample collection sites. Finally, the proportion was defined as the percentage of a particular ST present among all the samples in which the STs were identified.

**Statistical analyses.** Statistical analyses were performed using R v3.6.2 (47). A *P* value of <0.05 was considered to indicate statistical significance. The chi-square test was used to evaluate differences in *S. aureus* culture rates and proportions of STs among healthy skin, AD-NL, and AD-A. In addition, multivariate logistic regression analysis was performed to determine the clinical features associated with *S. aureus* infections. Logistic regression models were constructed using the R-glm function.

**Data availability.** This study generated sequencing data for 261 *S. aureus* isolates. All sequence read sets for this study are available from the DNA Data Bank of Japan database (http://www.ddbj.nig.ac.jp/) under BioProject accession no. PRJDB13052. In addition, the Sequence Read Archive (SRA) run accession numbers and associated metadata for each genome included in this study are listed in Table S2.

## SUPPLEMENTAL MATERIAL

Supplemental material is available online only.

**SUPPLEMENTAL FILE 1**, XLS file, 0.6 MB.
**SUPPLEMENTAL FILE 2**, PDF file, 0.8 MB.

## ACKNOWLEDGMENTS

We thank Hatsumi Maeo and Emi Numazaki for their assistance with this study. In addition, we thank Yuya Shimizu, Tomoki Yoshida, and Munechika Takaishi for their technical assistance with the bacterial culture and purification of genomic DNA.

This study was supported by the Japan Agency for Medical Research and Development (AMED) under the Advanced Research and Development Programs (grant no. JP21ek0410058, JP16gm1010001, JP22ek0410098, JP22ek0410079, 16gm1010001h0201, 17gm1010001h0202, 18gm1010001j0303, 19gm1010001j0304, 20gm1010001j0305, and 21gm1010001j0306) and the Research Program on Emerging and Re-emerging Infectious Diseases (grant no. JP21fk0108604, JP22fk0108604j0002, and 23fk0108604j0003). Furthermore, the research was supported by a university course research grant from the KOSE Cosmetology Research Foundation and the Naito Foundation. In addition, we received Health and Labor Sciences research grants (no. 21HA2009 and 21KA1004).

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
