## [Reviewer comments · Microbiology Spectrum]

Microbiology Spectrum

Comprehensive Genomic Characterization of *Staphylococcus aureus* Isolated from Atopic Dermatitis Patients in Japan: Correlations with Disease Severity, Eruption Type, and Anatomical Site

Shoko Obata, Junzo Hisatsune, Hiroshi Kawasaki, Ayano Fukushima-Nomura, Chika Arai, Kanako Masuda, Shoko Kutsuno, Yasuhisa Iwao, Motoyuki Sugai, Masayuki Amagai, and Keiji Tanese

Corresponding Author(s): Motoyuki Sugai, National Institute of Infectious Diseases

Review Timeline:

Submission Date:	December 22, 2022
Editorial Decision:	March 20, 2023
Revision Received:	June 8, 2023
Accepted:	June 12, 2023

Editor: Rosemary She

Reviewer(s): Disclosure of reviewer identity is with reference to reviewer comments included in decision letter(s). The following individuals involved in review of your submission have agreed to reveal their identity: Dennis Nurjadi (Reviewer #1)

Transaction Report:

DOI: <https://doi.org/10.1128/spectrum.05239-22>

March 20, 2023

Prof. Motoyuki Sugai
National Institute of Infectious Diseases
Antimicrobial Resistance Research Center
Aobacho 4-2-1
Higashimurayama-shi
Higashimurayama City 189-0002
Japan

Re: Spectrum05239-22 (Comprehensive Genomic Characterization of *Staphylococcus aureus* Isolated from Atopic Dermatitis Patients in Japan: Correlations with Disease Severity, Eruption Type, and Anatomical Site)

Dear Prof. Motoyuki Sugai:

Thank you for submitting your manuscript to Microbiology Spectrum. Your submission has been reviewed by two experts in the field and the consensus decision is Modifications. As you will see from the reviewer comments below, there was concern over the absence of data analysis on the clonality of staphylococcal isolates from the same patient, but it was felt that overall, with revision, the study would be of interest to the readership. When submitting the revised version of your paper, please provide (1) point-by-point responses to the issues raised by the reviewers as file type "Response to Reviewers," not in your cover letter, and (2) a PDF file that indicates the changes from the original submission (by highlighting or underlining the changes) as file type "Marked Up Manuscript - For Review Only". Please use this link to submit your revised manuscript - we strongly recommend that you submit your paper within the next 60 days or reach out to me. Detailed instructions on submitting your revised paper are below.

Link Not Available

Sincerely,

Rosemary She

Journals Department
Reviewer comments:

Reviewer #1 (Comments for the Author):

In the manuscript, the authors present the analysis of *S. aureus* isolates from atopic dermatitis patients in Japan. Altogether, 261 *S. aureus* isolates from 145 AD patients and 36 healthy controls were characterized by whole genome sequencing. The study premise is of interest since studies on *S. aureus* clonality in AD patients are scarce. Furthermore, *S. aureus* colonization and

infection are frequent in AD patients. The strength of the study is the study design, incorporating multi-site sampling and large surface area of sampling and a significant number of AD patients. The methods are described in detail, and sequenced were uploaded to a public database. The quality of written English is acceptable. The weaknesses of the study/manuscript are the data presentation, e.g. redundancy in figures and text. One of the main issues is the conclusion that AD skin is more likely to be inhabited by *S. aureus*. This notion, however, is not fully clear since only *S. aureus* positivity over all swabs taken was presented. For this claim, it would be necessary to present the positivity rate in AD patients (x/135) and not (x/579).

Moreover, this study could explore the aspect of clonality in a single patient (did all isolates in different localizations from patients inhabited by *S. aureus* belong to the same clone?). This analysis would add valuable information to this manuscript/study. The results presented can be more concise to highlight the main points.

Major comments

- There are too many figures/tables for the amount and depth of data presented. Some figures may not be necessary, Fig 1 (redundant to text), Fig 3-8 could be reduced. Many of these figures can be moved into supplementary data. The manuscript should be more focused on highlighting the main findings.
- Results lines 110-111: the denominator for percent positivity (% positivity) should be the number of patients in the first part of the results. This would give some impression on how many AD patients were colonized/infected and allow the readers to see how the *S. aureus* positivity in AD patients in Japan compared to other studies. Using the number of swabs as a denominator can distort the statistics. (Maybe I have missed this in the result since it was not mentioned in the beginning).
- Relating to the previous comment, did all patients have active lesions or just a proportion of all AD patients? This was unclear since the number of swabs was used as a denominator for the calculations. In my opinion, the culture rates are secondary to the %patients positive with *S. aureus* inhabitation and %patients with active lesions. Similarly, it was mentioned in lines 278-279 that the rate of *S. aureus* in AD patients is high. Please provide the % of AD patients with *S. aureus* colonization/infection (x/135).
- It was not mentioned whether the *S. aureus* isolates from a single patient are clonal (or not). This would be an interesting result for this specific study design.
- Discussion lines 274-276: AD is classically linked to a stronger Th2 response and more *S. aureus* colonization, whereas Psoriasis is linked with a stronger Th1 response and less *S. aureus* colonization. You may want to mention that the reason might be immunological/priming and not primarily strain variations/virulence profiles. There are numerous studies on this. Th1/Th17 response is important for *S. aureus* elimination in the skin.
- The text of the results is redundant and can be shortened (in combination with figures).
- Lines 309-310: strain abundance may be linked to molecular epidemiology of circulating strains in the region. Therefore, this argument is not convincing and is speculative. The concept of strain specificity for AD patients is not yet proven and established. The published data do not support this hypothesis since strain distribution is often linked to local/regional epidemiology.

Minor comments:

- Line 141, why is the ST untypable?
- Lines 155-156: 67 healthy individuals? Please check this; you may have meant AD patients with no active lesion. There are only 36 healthy controls declared in the study population section.
- Line 258: ethnic differences? Maybe "regional differences" is better?
- Lines 301-303: the meaning of the sentence is unclear
- Figures 3-8: legends are distorted.

Reviewer #2 (Comments for the Author):

The study prepared by Obata et al., described the characterization of *S. aureus* isolates collected from atopic dermatitis patients in Japan. This is a very interesting study and likely of interest for the readership of the journal. The study was well performed and I would have just a few minor comments described below.

1. On line 141, it's not clear what are the ST belonging to CC188. Also, if an ST is not typeable, how it can be associated with an CC. Perhaps,
2. This reviewer is not really sure the phylogenetic analysis, as it seems that the analysis ended up clustering the isolates into CC anyway. So, the analysis seems unnecessary. Please, clarify.
3. On line 152, can you define "superantigen". I am not sure this term is necessary, and authors can simply say "antigen genes".
4. On line 186, you can probably describe a little more detail about what "truncated" means. Would that be a premature stop codon?
5. The sentence in lines 202 - 204 reads as conclusion and it should be used in such section. The same for the sentence in lines 205 - 206.
6. I am not why the authors prepared an specific analysis/section for the comparison between isolates from Japan and Denmark, and in fact, characteristics of isolates from Europe and NA are also described. This seems a little off, and perhaps fits best in the discussion section as part of a comparison analysis with previously published data in the literature.

7. The initial paragraphs of the conclusion section repeat to much all results generated and I would suggest authors to minimize that and go straight to the discussion, as possible.

Staff Comments:

Preparing Revision Guidelines

Please return the manuscript within 60 days; if you cannot complete the modification within this time period, please contact me. If you do not wish to modify the manuscript and prefer to submit it to another journal, please notify me of your decision immediately so that the manuscript may be formally withdrawn from consideration by Microbiology Spectrum.

Jun 6th, 2023

Dr. Christina Cuomo
Editor in Chief
Microbiology Spectrum

Dear Dr. Cuomo,

We thank you and the reviewers for their enthusiasm and insightful comments on our manuscript (ID: Spectrum05239-22). We have comprehensively addressed all of the reviewers' comments and sincerely hope that the revised and improved manuscript is now deemed suitable for publication in *Microbiology Spectrum*. Please find below our detailed point-by-point responses to the reviewer comments. Revisions made to comply with the reviewer suggestions are highlighted in yellow in the revised manuscript.

We thank you for your assistance in the submission process and look forward to hearing from you.

Sincerely,

Keiji Tanese (tanese@a3.keio.jp)
Motoyuki Sugai (sugai@niid.go.jp)

Reviewer comments:

Reviewer #1 (Comments for the Author):

In the manuscript, the authors present the analysis of *S. aureus* isolates from atopic dermatitis patients in Japan. Altogether, 261 *S. aureus* isolates from 145 AD patients and 36 healthy controls were characterized by whole genome sequencing. The study premise is of interest since studies on *S. aureus* clonality in AD patients are scarce. Furthermore, *S. aureus* colonization and infection are frequent in AD patients. The strength of the study is the study design, incorporating multi-site sampling and large surface area of sampling and a significant number of AD patients. The methods are described in detail, and sequenced were uploaded to a public database. The quality of written English is acceptable. The weaknesses of the study/manuscript are the data presentation, e.g. redundancy in figures and text. One of the main issues is the conclusion that AD skin is more likely to be inhabited by *S. aureus*. This notion, however, is not fully clear since only *S. aureus* positivity over all swabs taken was presented. For this claim, it would be necessary to present the positivity rate in AD patients (x/135) and not (x/579).

Moreover, this study could explore the aspect of clonality in a single patient (did all isolates in different localizations from patients inhabited by *S. aureus* belong to the same clone?). This analysis would add valuable information to this manuscript/study. The results presented can be more concise to highlight the main points.

Two comments raised by the reviewer#1 in two paragraphs were a) present the positivity rate in AD patient (x/135) and b) analyze clonality of *S. aureus* inhabited in different localization of the patients. Response to a) and b) were described in responses below 1 and 3

Major comments

1 There are too many figures/tables for the amount and depth of data presented. Some figures may not be necessary, Fig 1 (redundant to text), Fig 3-8 could be reduced. Many of these figures can be moved into supplementary data. The manuscript should be more focused on highlighting the main findings.

Response: In accordance with this suggestion, we have dropped Fig. 1A, and Fig. 1B–1D have all been moved to the supplemental figures (Fig. S2A-C).

Figure 3 (previous version): We have deleted Fig. 3A count data.

Figure 4 (previous version): We have deleted Fig. 4A count data.

Figure 5 (previous version): We have deleted Fig. 5A count data.

Figure 6 (previous version): We have deleted Fig. 6A count data.

Figures 7 and 8 (previous version): (A) and (B) were reformed to heatmaps.

2 Results lines 110-111: the denominator for percent positivity (% positivity) should be the number of patients in the first part of the results. This would give some impression on how many AD patients were colonized/infected and allow the readers to see how the *S. aureus* positivity in AD patients in Japan compared to other studies. Using the number of swabs as a denominator can distort the statistics. (Maybe I have missed this in the result since it was not mentioned in the beginning).

Response: Results line 119 (revised manuscript): We had incorrectly written this sentence. We fully agree with your suggestions. The denominator for the positivity rate (%positivity) was corrected from the number of swabs to the number of patients.

3 Relating to the previous comment, did all patients have active lesions or just a proportion of all AD patients? This was unclear since the number of swabs was used as a denominator for the calculations. In my opinion, the culture rates are secondary to the %patients positive with *S. aureus* inhabitation and %patients with active lesions. Similarly, it was mentioned in lines 278-279 that the rate of *S. aureus* in AD patients is high. Please provide the % of AD patients with *S. aureus* colonization/infection (x/135).

Response: As pointed out, we compared, with previous studies, the rates of colonization (i.e. non-active) and infection (i.e. active) per AD patient as a population.

We have revised the text accordingly (revised lines: 264-266).

4 It was not mentioned whether the *S. aureus* isolates from a single patient are clonal (or not). This would be an interesting result for this specific study design.

Response: We have calculated the percentage of single or multiple clones from each AD patient and have included this in lines 266-268.

5 Discussion lines 274-276: AD is classically linked to a stronger Th2 response and more *S. aureus* colonization, whereas Psoriasis is linked with a stronger Th1 response and less *S. aureus* colonization. You may want to mention that the reason might be immunological/priming and not primarily strain variations/virulence profiles. There are numerous studies on this. Th1/Th17 response is important for *S. aureus* elimination in the skin.

Response: The points raised have been added to lines 261-263.

6 The text of the results is redundant and can be shortened (in combination with figures).

Response: We have shortened some of the text in the Results (indicated via the strike-through feature in the revised manuscript).

7 Lines 309-310: strain abundance may be linked to molecular epidemiology of circulating strains in the region. Therefore, this argument is not convincing and is speculative. The concept of strain specificity for AD patients is not yet proven and established. The published data do not support this hypothesis since strain distribution is often linked to local/regional epidemiology.

Response: The sentence in lines 309-310 of the original version has been deleted, indicated via the strike-through feature.

Minor comments:

8 Line 141, why is the ST untypable?

Response: For the untypable ST, the seven allele numbers were *arcC_3*, *aroE_1** (novel allele), *glpF_1*, *gmk_8*, *pta_1*, *tpi_1*, and *yqiL_1*. Based on this allele pattern and the eBURST analysis, it was assumed to belong to CC188. For brevity in the text, "ST untypable" has been deleted (revised version line 148, indicated via the strike-through feature) and replaced by "the untyped STs".

9 Lines 155-156: 67 healthy individuals? Please check this; you may have meant AD patients with no active lesion. There are only 36 healthy controls declared in the study population section.

Response: Line 115 and 120-125 (revised manuscript): We have added a sentence clarifying this.

10 Line 258: ethnic differences? Maybe "regional differences" is better?

Response: Line 351 (revised manuscript): We have revised this phrase to "regional differences," as suggested.

11 Lines 301-303: the meaning of the sentence is unclear

Response: We have deleted this section since it is unclear.

12 Figures 3-8: legends are distorted.

Response: We have revised the distorted legends in Figures 3-8.

Reviewer #2 (Comments for the Author):

The study prepared by Obata et al., described the characterization of *S. aureus* isolates collected from atopic dermatitis patients in Japan. This is a very interesting study and likely of interest for the readership of the journal. The study was well performed and I would have just a few minor comments described below.

1. On line 141, it's not clear what are the ST belonging to CC188. Also, if an ST is not typeable, how it can be associated with an CC. Perhaps,

Response: For the untypable ST, the seven alleles were *arcC_3*, *aroE_1** (novel allele), *glpF_1*, *gmk_8*, *pta_1*, *tpi_1*, and *yqiL_1*. Based on this allele pattern and the eBURST analysis, it was assumed to belong to CC188. Line 148-149 (revised manuscript); we have rephrased as “ST188, ST1533, and untyped ST belonging to clonal complex (CC) 188.”

2. This reviewer is not really sure the phylogenetic analysis, as it seems that the analysis ended up clustering the isolates into CC anyway. So, the analysis seems unnecessary. Please, clarify.

Response: We have investigated the phylogenetic relationships using core genome SNP analysis based on the whole genome sequences, as shown in Figure 2. The grouping of clusters by SC was based on genome similarity. This figure depicts not only the results of clustering (the clusters in the phylogenetic tree were closely correlated with the CC group) but also the similarity between clusters (and STs). Therefore, we can discuss the similarity of STs (CCs) in conjunction with sampling metadata to make the explanation easy and concise.

3. On line 152, can you define "superantigen". I am not sure this term is necessary, and authors can simply say "antigen genes".

Response: Superantigens are proteins produced by bacteria that are known to induce an immune system reaction. Superantigens can stimulate the immune system to cause abnormal immune reactions. *S. aureus* produces superantigenic toxins such as TSST-1 and many *Staphylococcal* enterotoxins. For example, TSST-1 strongly binds to specific receptors on T cells and activates them, causing abnormal immune reactions. This can cause symptoms such as fever, rash, and low blood pressure. Therefore, it is not common to simply refer to them as "antigen genes."

4. On line 186, you can probably describe a little more detail about what "truncated" means. Would that be a premature stop codon?

Response: Line 189-190 (revised manuscript): We have included “due to a recombination event between the direct repeats within these genes,” in the revised manuscript.

5. The sentence in lines 202 - 204 reads as conclusion and it should be used in such section. The same for the sentence in lines 205 - 206.

Response: Line 291-295 (revised manuscript): We have moved this to the Discussion, as suggested.

6. I am not why the authors prepared an specific analysis/section for the comparison between

isolates from Japan and Denmark, and in fact, characteristics of isolates from Europe and NA are also described. This seems a little off, and perhaps fits best in the discussion section as part of a comparison analysis with previously published data in the literature.

Response: Line 348-368 (revised manuscript): We have moved this to the Discussion, as suggested.

7. The initial paragraphs of the conclusion section repeat to much all results generated and I would suggest authors to minimize that and go straight to the discussion, as possible.

Response: Line 257-259 (revised manuscript): We have revised the first paragraph of the Discussion to minimize repetition and make it more concise, as suggested.

June 12, 2023

Prof. Motoyuki Sugai
National Institute of Infectious Diseases
Antimicrobial Resistance Research Center
Aobacho 4-2-1
Higashimurayama-shi
Higashimurayama City 189-0002
Japan

Re: Spectrum05239-22R1 (Comprehensive Genomic Characterization of *Staphylococcus aureus* Isolated from Atopic Dermatitis Patients in Japan: Correlations with Disease Severity, Eruption Type, and Anatomical Site)

Dear Prof. Motoyuki Sugai:

Your manuscript has been accepted, and I am forwarding it to the ASM Journals Department for publication. You will be notified when your proofs are ready to be viewed.

Sincerely,

Rosemary She
Editor, Microbiology Spectrum
